# A Swiss-Roll-Type Methanol Mini-Steam Reformer for Hydrogen Generation with High Efficiency and Long-Term Durability

**DOI:** 10.3390/mi14101845

**Published:** 2023-09-27

**Authors:** Fan-Gang Tseng, Wei-Cheng Chiu, Po-Jung Huang

**Affiliations:** Department of Engineering and System Science, National Tsing Hua University, Hshinchu 300, Taiwan; tigertank87@yahoo.com (W.-C.C.); xx0912896885@gapp.nthu.edu.tw (P.-J.H.)

**Keywords:** methanol reformer, SRM, Swiss-roll reactor, hydrogen generation, GHSV, durable reformer, long-term stability

## Abstract

This paper proposes a Swiss-roll-type mini-reformer employing a copper–zinc catalyst for high-efficient SRM process. Although the commercially available copper–zinc catalysts commonly used in cylindrical-type reformers provide decent conversion rates in the short term, their long-term durability still requires improvement, mainly due to temperature variations in the reformer, catalyst loading, and thermal sintering issues. This Swiss-roll-shaped mini-reformer is designed to improve thermal energy preservation/temperature uniformity by using dual spiral channels to improve the long-term durability while maintaining methanol-reforming efficiency. It was fabricated on a copper plate that was 80 mm wide, 80 mm long, and 4 mm high with spiral channels that were 2 mm deep, 4 mm wide, and 350 mm long. To optimize the design and reformer operation, the catalyst porosity, gas hourly speed velocity (GHSV), operation temperature, and fuel feeding rate are investigated. Swiss-roll-type reformers may require higher driving pressures but can provide better thermal energy preservation and temperature uniformity, posing a higher conversion rate for the same amount of catalyst when compared with other geometries. By carefully adjusting the catalyst bed porosity, locations, and catalyst loading amount as well as other conditions, an optimized gas hourly space velocity (GHSV) can be obtained (14,580 mL/g·h) and lead to not only a high conversion rate (96%) and low carbon monoxide generation rate (0.98%) but also a better long-term durability (decay from 96% to 88.12% after 60 h operation time) for SRM processes. The decay rate, 0.13%/h, after 60 h of operation, is five-folds lower than that (0.67%/h, 0.134%/h) of a commercial cylindrical-type fixed-bed reactor with a commercial catalyst.

## 1. Introduction

Fossil fuel is still an important energy source and widely utilized nowadays. Especially in transportation, it is responsible for 60% of the global oil demand and 30% of the world’s total delivered energy [1]. As air pollution, global warming, and the associated climate change become more urgent issues due to the increasing usage of fossil fuels, a replacement has been considered for a long time, and hydrogen could be a good choice [2]. However, the impact will be high on industry and the economy without careful planning of the replacement. Among various new energy sources, hydrogen has been considered as a clean and countable energy source to replace fossil fuel for not only reducing much of the environment impact but also providing significantly higher efficiencies of energy utilizations. Hydrogen can provide a wide range (22–60%) of efficiency in automobile internal combustion engines since hydrogen combustion can be applied either in lean-fuel/air or rich-mixture environments, while fossil-fuel combustion can only rely on rich mixtures [3,4]. Hydrogen can also provide electricity through fuel cells with much greater efficiencies at 65–70% [5], while the efficiencies of heat power plants are between 35 and 38%. On the other hand, when integrated with an electrical motor, fuel cells can also be used in electrical transport vehicles with higher efficiencies (40–60%) [6] than those (11–25%) [7] of gasoline-driven motors.

Despite the many advantages of using hydrogen as an energy source, the production, storage, and transportation of hydrogen are still very troublesome and not cost-effective nowadays [8,9]. Therefore, methanol is considered a good alternative to provide hydrogen after reforming processes for usage in integrated hydrogen-based power systems For instance, under standard conditions, methanol’s volume energy density (4.82 kWh/L) is higher than that of hydrogen (0.0028 kWh/L) (methanol and hydrogen at normal temperature and pressure). Furthermore, the requirements for the storage of methanol are much easier when compared to hydrogen. Methanol has a high hydrogen-to-carbon ratio when compared to other hydrocarbons, and its sulfur content and reforming temperature are much lower, which is important for fuel cell utilization [10]. The hydrogen stored in methanol is considered stable, not flammable, and has a high energy density when compared to pressure-vessel-stored hydrogen [11]. Among various methanol reforming processes, the steam reforming of methanol (SRM) reaction has the advantages of a high ratio of hydrogen generation and low CO generation [12]. In SRM processes, most research has employed a large amount of catalyst with low flow rates to ensure low CO generation and high conversion rates to minimize damage to the fuel cell catalyst [13,14]. However, the long-term durability of a catalyst is still an issue due to the fact that copper catalysts are especially susceptible to careless temperature excursions that lead to the excessive sintering of the copper crystallites and an irreversible loss of activity [15]. The primary method of large-scale hydrogen production is steam methane reforming (SMR) in the packed-bed reactors filled with catalysts. In these reactors, heat is mostly supplied through the reactor wall to compensate for the endothermic effect of SMR reactions [16]. To ensure the efficient operation of the hydrogen production reactor, synchronous heat generation and consumption within the channels are crucial. Localized hotspots must be eliminated to ensure the full utilization of the catalyst in the reformer channels [17].

Conventionally, there are more than four kinds of catalysts developed for commercial steam reforming processes, basically containing Cu, Zn, Al, and their oxides [18]. The conversion rate and CO mole fraction would be different for different ratios of Cu, Zn, and Al and the methods to produce the commercial catalysts [19]. The catalyst that used in this study is Cu/ZnO/Al_2_O_3_, BASF K3-110, one of the most adopted commercial catalyst for steam reforming processes [18,20,21]. To solve the long-term durability issues, the chemical, thermal, and mechanical aspects of catalyst deactivations can be solved, respectively, when dealing with catalyst weight, pore size, process temperature optimization. and spatially dependent or a temporally dependent mechanisms. Since the current device employs a commercially available catalyst, this study will mainly focus on the thermal and mechanical issues for improving the long-term durability of catalyst operations in a more thermally/mechanically stabled reformer. The effect of operation temperature is one of the most important causes of catalyst sintering [22,23], which would not only decrease the productivity of the product but also lower the catalyst life time due to a reduction in the catalyst active surface area [23]. On the other hand, the catalyst may also encounter mechanical deactivations by being crushed into finer particles after long-term operation/oxidation processes, which cause either fluid channeling through the catalyst’s empty regions without a reaction or a complete blocking of the flow due to fine-particle stacking. Flow resistance can thus be changed at different channel locations, leading to uneven fluid flow for wave-front poisoning and/or coking at an accelerated rate [23]. These stagnant regions can develop into “hot spots” within the catalyst mass where temperatures become high enough to induce crystalline phase changes [19,24].

Therefore, in this study, a Swiss-roll-shaped reformer is proposed to help better temperature preservation/distribution [25] in dual spiral channels to improve the long-term durability while maintaining methanol-reforming efficiency. To optimize the performance, both the GHSV and catalyst porosity are investigated. The GHSV is defined as the ratio of the gas flow rate under standard conditions to the volume of the catalyst bed for a reactor, which is the reciprocal of the chemical reaction time [26]. The dimension of the GHSV is *q*/(*c·t*), while *q* is the reactant gas volume flow per hour, *c* is the weight of catalyst loading, and *t* is the retention time of the reactant gas [27]. A faster flow velocity caused by a higher GHSV results in a shorter contact time between the reactants and the catalyst, which leads to a lower conversion rate but higher product (hydrogen) generation [28]. Choi et al. found that, when operated at high a GHSV and temperature, more CO generation and a higher conversion rate can be observed in a water–gas-shift reaction on the Cu/ZnO/Al_2_O_3_ catalyst [29]. Karim et al. also depicted that the catalyst activity will be deteriorated due to a large temperature gradient in the reactor, which is generated by increasing the reactor size and GHSV for the Cu/ZnO/Al_2_O_3_ catalyst [19]. In terms of catalyst porosity, Herdem et al. found that increasing the porosity would increase the conversion rate under the simulation condition of a heat flux of 1000 W/m^2^ and a porosity of 0.4 [30]. Hui An et al. observed that a coiled-type channel design among eight different types of reactors can offer the highest conversion rate through the optimization of temperature, pressure, and fuel utilization altogether [21]. After considering all the previous works, it is the ultimate goal of this study to find the optimum GHSV and catalyst porosity for better long-term durability while not losing too much hydrogen generation/conversion rate in the Swiss-roll-type methanol reformer. The efficiency of the steam methane reforming process is influenced by several parameters. These parameters can be categorized into two main groups: the operational factors, including temperature, pressure, and the inlet steam-to-methane ratio; and the reformer design factors, including the reformer geometry, type of catalysis, catalytic shape, and residence time. To establish the correlation between these parameters and the efficiency of the steam methane reforming process, a combination of experimental research and numerical simulation is employed. Modern numerical techniques such as CFD modeling are utilized for a more in-depth exploration of the phenomena within the steam methane reforming process [31]. Improving the efficiency of the reforming process necessitates a comprehensive investigation of both the chemical equilibrium and chemical kinetics within the reactor. The equilibrium values are typically determined through a thermodynamic analysis, while the reaction kinetic values are commonly acquired from experiments and CFD modeling [16]. The most prevalent approach for the steam methane reforming (SMR) process involves chemical reactions taking place near porous catalysts with external heat input. External heat is typically delivered to the reactor walls, which subsequently transfer the heat to the reaction zone via various methods, including fossil fuel combustion, solar heating, Joule heating, and waste-heat recovery. The actual reaction transpires within packed-bed-containing catalysts of diverse shapes and materials. However, this heat exchange method comes with a significant drawback associated with the reactor wall material. The reactor wall experiences high temperatures due to the heat supply zone’s exhaust flue gases and uneven temperature distribution from the strongly endothermic reactions of SMR. Consequently, the reactor wall is subjected to considerable thermal stress, potentially leading to material degradation. Furthermore, the transfer of heat between the exhaust gases and reaction mixture is impeded by the thermal resistance posed by the reactor wall [16].

## 2. Design of the Swiss Roll Reformer

The cylindrical type is the most commercially adopted reactor, which consists of a tube containing a catalyst inside the hollow area of the tube, as shown in Figure 1a; the channels have a length of 350 mm, a width of 4 mm, and a depth of 2 mm. However, the Swiss-roll-type reformer can have a more uniform temperature distribution by swirling the tube to reduce the distance between the tube inlet and outlet for promoting the heat transfer, as shown in Figure 1b. The reaction volume, channel length, cross-sectional area, aspect ratio, and material of the reaction channels are designed the same in this study for the tube-type and Swiss-roll-type reformers for comparison. The simulation results of the pressure, velocity, and temperature distributions for these two types of reformers are illustrated in Figure 2 and were obtained using ANSYS 2023 R2 software, and the differences are tabulated in Appendix A. The details of the boundary conditions, geometries, and the meshes are listed in Appendix A and Figure 2.

Mathematical formulation Governing equations

Continuity:(1)$$\frac{\partial \rho u}{\partial x}+\frac{\partial \rho v}{\partial y}=0$$

Momentum:(2)$$\left(u\frac{\partial \rho u}{\partial x}+v\frac{\partial \rho v}{\partial y}\right)=-\frac{\partial \rho }{\partial x}+(\frac{\partial }{\partial x}(\mu \frac{\partial u}{\partial x})+\frac{\partial }{\partial y}\left(\mu \frac{\partial u}{\partial y}\right))$$
(3)$$\left(u\frac{\partial \rho v}{\partial x}+v\frac{\partial \rho v}{\partial y}\right)=-\frac{\partial \rho }{\partial y}+(\frac{\partial }{\partial x}(\mu \frac{\partial v}{\partial x})+\frac{\partial }{\partial y}\left(\mu \frac{\partial v}{\partial y}\right))$$

Energy:(4)$$(u\frac{\partial \rho C_{p}T}{\partial x}+v\frac{\partial \rho C_{p}T}{\partial y})=\frac{\partial }{\partial x}\left(\lambda \frac{\partial T}{\partial x}\right)+\frac{\partial }{\partial y}\left(\lambda \frac{\partial T}{\partial y}\right)$$

Species:$$u\frac{\partial \rho w_{i}}{\partial x}+v\frac{\partial \rho w_{i}}{\partial y}=\frac{\partial }{\partial y}\left(D_{i}\frac{\partial^{2}\rho w_{i}}{\partial x^{2}}\right)+\frac{\partial }{\partial y}\left(D_{i}\frac{\partial^{2}\rho w_{i}}{\partial y^{2}}\right)$$

Diffusion equation:(5)$$D_{i}=\frac{\lambda }{\rho Le_{i}C_{p}}$$
(6)$$\begin{array}{l} \mu =\sum_{i=1}^{n}\;w_{i}\mu_{i}\;\\ C_{p}=\sum_{i=1}^{n}\;w_{i}C_{p,i}\\ \lambda =\sum_{i=1}^{n}\;w_{i}\lambda_{i} \end{array}$$

When comparing the pressure drops, one can find that the pressure drop (140.542 Pa) of the tube type is 10% lower than that (152.717 Pa) of the Swiss-roll reformer (Figure 2a,d), attributing to several changes in the flow directions in the Swiss-roll channel, which can be found by comparing Figure 2b,e. In addition, the tube-type reformer shows a more uniform velocity distribution, while for the Swiss-roll-type reformer, there are some velocity fluctuations at the corners of the channel (Figure 2d,e).The velocity difference (0.262 m/s) of the Swiss-roll type is thus approximately 1.5 higher than that (0.181 m/s) of the tube type. The change in fluid directions would lead to a difference in velocity and an increase in wall shear, which would increase the pressure drop and pumping power in the Swiss roll reformer. When comparing the temperatures, one can find not only that the temperature variation (0.5 K) of Swiss roll reformer is much less than that (1.8 K) of tube reformer, but also that the average temperature is 3 k higher. Thus, a better conversion rate and catalyst sintering situation can be anticipated for the Swiss-roll channel design based on its better performance and long-term stability by trading off a little higher (10%) pumping power. To make a fair comparison between the traditional straight-type fixed-bed reformer and the Swiss-roll one, we arranged all the important specs similarly, including the total length of channel (350 mm) and cross-sectional area (4 mm–2 mm). The simulation was conducted by using ANSYS software 2023 R2. The simulation results are shown in Figure 2, and the simulation parameters are listed in Table 1 and Table 2. Comparing Figure 2a,d, one can find that the pressure drop (140.542 Pa) of the tube-type reformer is 10% lower than that (152.717 Pa) of the Swiss-roll reformer. Figure 2d shows a more uniform velocity distribution in the tube-type reformer, while from Figure 2e, some velocity fluctuations can be found at the corners of the channel in the Swiss-roll one. When comparing the temperature distributions in Figure 2c,f, one can find not only that the temperature variation (0.5 K) of Swiss roll reformer is much less than that (1.8 K) of tube one, but also the average temperature is 3 k higher.

## 3. Materials and Methods

### 3.1. Experiment Setup of SRM Reformer

The design schematic and fabricated Swiss-roll reformer is shown in Figure 3a. It was made of copper and was 80 mm wide, 80 mm long, and 4 mm high. The reformer channel was 2 mm deep, 4 mm wide, and 354 mm long. A heating belt was wrapped around the reformer as a heat source to control the reformer temperature from room temperature to the reaction temperature. Furthermore, a heat-resistant gasket was arranged between the flow channel and the cover plate to avoid gas leakage. 

The assembly process and experiment setup of the reformer system is depicted in Figure 3b. First, the methanol reforming catalyst, Cu/ZnO/Al_2_O_3_, was grinded to a size of 420–590 μm and loaded into the reformer channel, as well as fixed with copper mesh at the front and end of the catalyst. When the reformer was heated to the designed temperature (230 °C–250 °C) using a heating belt, premixed DI water/methanol fuel was fed into a vaporization chamber via a peristaltic pump before entering the reformer. Finally, the conversion rate and CO mole fraction were measured by a Gas Chromatography (GC) machine.

### 3.2. Optimization of Catalyst Porosity 

A commercial catalyst composition (wt%) of 40% CuO, 40% ZnO, and 20% Al_2_O_3_ (BASF K3-110, BASF company, Ludwigshafen, Germany) [18] with a weight of 1 g was employed for the SRM reformer. A methanol solution with a steam-to-carbon (S/C) ratio of 1.2 was fed into the reformer at 0.33 mL/min. To adjust the porosity and flow resistance of the catalyst for the reaction, various filling volumes (80 cm^3^–112 cm^3^) of the catalyst in the reformer were used to balance between the heat transfer performance (at lower porosity) and flow resistance (at higher porosity) to optimize fuel residence time and prevent a reverse water–gas-shift (WGS) reaction for improving overall performance. Thus, the optimized porosity can be determined and employed for optimizing GHSV.

### 3.3. Optimization of Reformer GHSV 

Since GHSV is the ratio of the gas flow rate to the volume of the catalyst bed for a reactor, it is important to adjusted the GHSV value until the performance of the reformer reaches the optimized conditions. Faster flow velocity caused by higher GHSV results in a shorter contact time between the reactants, which leads to the catalyst having a lower conversion rate but higher product (hydrogen) generation [26]. By changing the loading amount and porosity of catalyst to control the GHSV parameters, the residence time of the fuel gas in the catalyst bed can greatly affect the methanol conversion efficiency. In this test, the GHSV parameters were sorted according to the porosity test data combined with the catalyst loading amount; thus, the optimized methanol conversion rate can be found based on these two parameters. The carbon monoxide concentration and hydrogen production were also recorded to ensure the optimized conditions for SRM.

### 3.4. Long-Term Durability Test

The catalyst bed with the optimal parameters of GHSV and porosity is then installed in the flow channel of the reformer for the long-term test. The operating temperature was controlled between 230 °C and 250 °C, and the reformer was tested for 60 h. The methanol conversion rate was recorded as an indicator of performance reduction.

## 4. Experiment Results

### 4.1. Catalyst Porosity Test

The optimized porosity of the Swiss-roll reformer was assessed by verifying the methanol conversion rate and CO mole fraction at different temperatures and inlet flow rates. The porosity was set up by arranging 1 g of the catalyst into different lengths/volumes in the reaction channels. As shown in Figure 4 and Figure 5, the methanol conversion rate increases with a higher temperature, lower inlet flow rate, and smaller porosity. Decreasing the inlet flow rate and porosity leads to a longer gas residence time, while increasing the temperature to induce a faster reaction leads to a higher conversion rate. However, a longer residence time and higher temperature would both cause the catalyst bed temperature to overheat and result in a reverse WGS (rWGS) reaction and a higher CO generation. As shown in Figure 4, the CO mole fraction increases with an increase in temperature and with a decrease in the inlet fuel flow rate. Since reducing the CO mole fraction below 1% is important for phosphorous acid fuel cell (PAFC) applications [29] in the future, the operating temperature should be set at 240 °C, with a porosity of 40%, and an inlet flow rate of 0.33 mL/min for optimized performance with a decent conversion rate (87%) and reasonable CO generation fraction (0.9%), as shown in Figure 4. It is noted that when the inlet flow rate decreases from 0.33 mL/min to 0.17 mL/min, the conversion rate gradually becomes flat and stable. Once the optimized porosity of 40% is obtained, then the overall performance of the Swiss-roll reformer can be determined by the GHSV in the next section.

### 4.2. GHSV Test

To obtain the best performance of the current Swiss-roll reformer as well as the highest methanol conversion and hydrogen production rate while keeping a reasonable CO fraction, the gas residence time (GHSV) will be considered. The relationship between catalyst weight and GHSV is shown in Appendix A. In this regard, the commercial catalyst, with a weight ranging from 1 g to 2.5 g, was filled into the flow channels of the reformers, with a similar porosity of 40%, and methanol solution with a fixed S/C ratio of 1.2 was used as the fuel. According to the porosity test in the previous section, a high conversion rate could be obtained when the operating temperature was set between 240 °C and 250 °C. Therefore, three operating temperatures of 230 °C, 240 °C, and 250 °C were selected for testing in this session. The related methanol conversion rates and CO mole fractions for different GHSV gas residence times were also investigated. As shown in Figure 5, when the temperature was raised to 240 °C and 250 °C, both reformers could reach a similarly high methanol conversion (96–98%) and hydrogen production (330–380 sccm) rate. The highest value for the methanol conversion and hydrogen production rate at all temperatures can be found at a GHSV value equal to 14,583 mL/g·h and a fuel feeding rate of 0.33 mL/min. In addition, the measured methanol conversion rates and hydrogen production rates for other GHSV values were lower than at a GHSV of 14,583 mL/g·h, as shown in Figure 5a,c. Therefore, under these experimental conditions, we have determined that the optimal GHSV parameter is 14,583 mL/g·h. The optimized catalyst amount would be 2.0 g, as illustrated in Figure 5 and Table 2. It is noticeable that the reformer performance would be reduced when the catalyst weight is further increased to 2.25 and 2.5 g. To keep the CO mole fraction lower than 1% for its further application to high-temperature Proton Exchange Membrane Fuel Cells (PEMFCs), such as PAFC [32], the operation temperature of the Swiss-roll reformer is preferentially set at 240 °C while keeping a similarly high methanol conversion (96%) and hydrogen generation (330 sccm) rate at a low CO production (0.98%), as shown in Figure 5. 

### 4.3. Long-Term Durability Test of Swiss-Roll Reformer

In the long-term durability test, combined with the test data from the previous two tests, the Swiss-roll reformers with optimal porosity (40%) and catalyst weight (2 g), but three different working temperatures (230, 240, and 250 °C) and GHSVs (12,963, 14,583, and 16,667 mL/g·h) were selected for comparison. According to the GHSV test data, a gas feeding rate of 14,583 (mL/(g·h)) provides the best performance, so it is selected to serve as a fixed testing parameter for temperature variation testing. On the other hand, 240 °C is selected as the optimal temperature for GHSV variation testing.

When all the GHSV parameters were fixed at 14,583 (mL/(g·h)) and the operating temperature varied from 230 to 250 °C for 60 h of operation, as shown in Figure 6 and Appendix A, the 240 °C case demonstrated a decent low decay rate (0.1322%/h) similar to that (0.1297%/h) of the 230 °C one, while it maintained a good final conversion rate (88.12%) close to that (88.66%) of the 250 °C one, which can be recognized as the optimal selection when considering the long-term durability.

The long-term test results of various GHSVs with a fixed operating temperature of 240 °C are shown in Figure 7. The methanol conversion rate at a GHSV of 16,667 (mL/(g·h)) is lower than that of the other two parameters early on in the long-term durability test due to the low catalyst-loading weight. With the reaction time running into 60 h at the end of the test, it was observed that either the long-term durability or the final conversion rate was still optimal for a GHSV of 14,583 (mL/g·h). It is suggested that a high GHSV would shorten the gas residence time for the fuel gas, leaving the catalyst too little time for the complete reaction to deteriorate the conversion rate, while a low GHSV also reduces the conversion rate due to the gas residence time being too long to induce a reverse WGS reaction. As a result, finding the optimal GHSV is critical and effective in designing a decent reformer with high performance, not only in terms of methanol conversion/hydrogen production rate but also in terms of long-term durability. 

## 5. Discussion

Based on all the results from the previous sections, the optimal performance of the 5 cm × 5 cm Swiss-roll SRM reformer can be summarized as a 2 g loading weight of the catalyst with a 40% porosity, a 14,583 mL/g·h GHSV, a 240 °C operation temperature, and feeding with a 0.33 mL /min methanol solution of S/C 1.2. A high conversion rate (96%), low CO mole fraction (0.98%), high hydrogen productivity (350 sccm), and decent long-term durability (decay rate of 0.1322%/h) are obtained when compared to those obtained (calculate decay rate of 0.67%/h) by Conant, T. et al. [33] for a cylindrical-type reformer utilizing the same catalyst, BASF F13456, and operating at a 16,667 mL/g·h GHSV (equal to 27 kg·s/mol). Furthermore, when compared with the other reformer, equipped with a common commercial catalyst (G66B, Sud Chemie Catalyst Inc., Tokyo, Japan by G. Huang et al.) [34] and operated at a GHSV of 21,152 mL/g·h (g_fed_ = 3.5 mL/h, g_cat_ = 0.22 g), a similar decay rate (0.1322%/h) but very low conversion rate (45%–30%) was observed. The methanol conversion rate of the Swiss-roll reformer dropped to 88.5%, while for the cylindrical-type reformer, it dropped to 55% in [29] after 60 h of long-term operation. As a result, the Swiss-roll-type SRM reformer can provide better performance, not only in terms of methanol conversion/hydrogen production but also in terms of long-term durability for a presumably more uniform temperature distribution and better thermal energy preservation than the other types, such as the fractal-channel-type [35] and shell- and tube-type [36] reformers, when carefully optimizing the catalyst porosity, GHSV, and operation temperatures. In this study, our approach to enhancing the overall performance of the methanol steam reforming reaction does not involve direct catalyst improvement. Instead, we focus on modifying the catalyst bed configuration to improve the temperature uniformity and pressure drop across the catalyst. The results demonstrate that, compared to the conventional cylindrical-type reformer under the same flow channel conditions, the Swiss-roll reformer with an altered geometric design exhibits superior performance.

## 6. Conclusions

This paper has illustrated a 5 cm × 5 cm Swiss-roll-type reformer employing a copper–zinc catalyst to improve the long-term durability of traditional cylindrical-type reformers from temperature variation, catalyst loading, and thermal preservation issues for high-efficient SRM processes. This Swiss-roll-shaped reformer employs a dual spiral channel design to help with thermal energy preservation/uniform temperature distribution in the channels to improve long-term durability while maintaining methanol reforming efficiency. To optimize the design and reformer operation, the catalyst porosity, gas hourly speed velocity (GHSV), operation temperature, and fuel feeding rate were investigated. The optimal design and operation conditions included a 2 g catalyst loading with a 40% porosity, a 14,583 mL/g·h GHSV, a 240 °C operation temperature, and feeding with a 0.33 mL/min methanol solution of S/C 1.2 for the Swiss-roll reformer. A high conversion rate (96%), low CO mole fraction (0.98%), high hydrogen productivity (350 sccm), and decent long-term durability (decay rate of 0.1322%/h) were also obtained. The decay rate, 0.13%/h, after 60 h of operation, was five-folds lower than that (0.67%/h) of a commercial cylindrical-type fixed-bed reactor loaded with a similar commercial catalyst. The current Swiss-roll reformer, as the first design model optimized in this study, can be further expanded into a larger version with in-parallel spiral channels designed to generate hydrogen for a 1 kW fuel cell system in the future.

## Figures and Tables

**Figure 1 micromachines-14-01845-f001:**
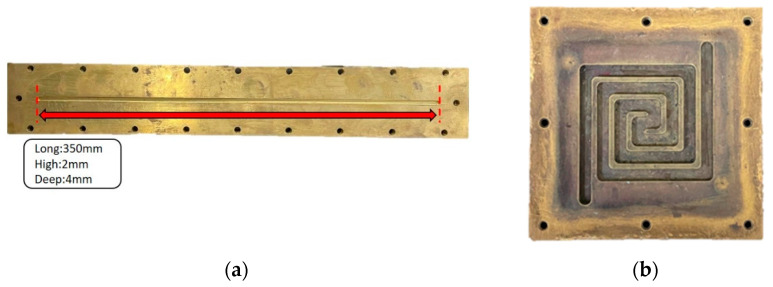
(**a**) Tube type; (**b**) Swiss roll type.

**Figure 2 micromachines-14-01845-f002:**
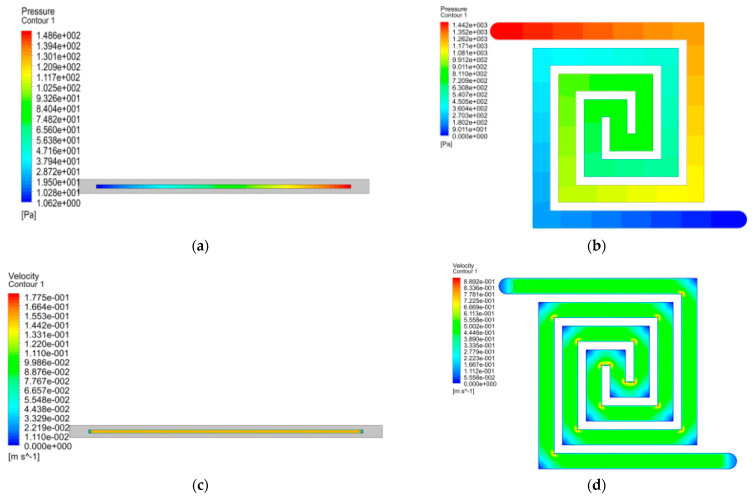
Pressure, velocity, and temperature distribution of (**a**–**c**) tube-type and (**d**–**f**) Swiss-roll-type reformer.

**Figure 3 micromachines-14-01845-f003:**
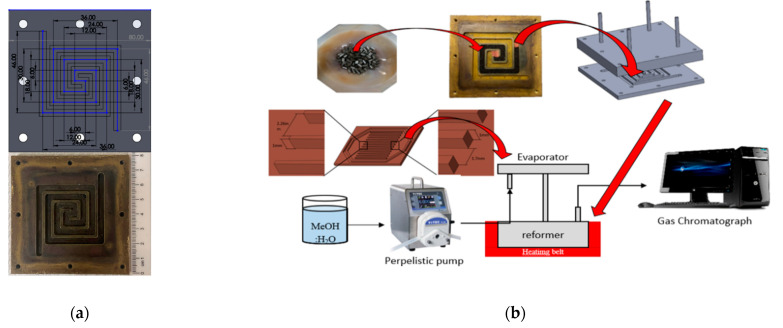
Swiss-roll reformer: (**a**) design schematic and fabricated channel, and (**b**) assembly and experiment setup.

**Figure 4 micromachines-14-01845-f004:**
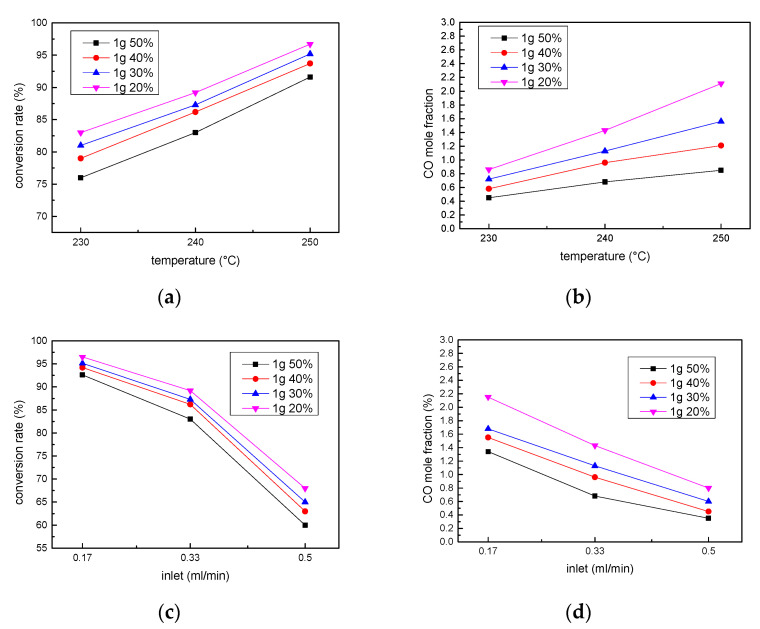
The methanol conversion rates and CO mole fractions of Swiss-roll reformer with different porosities at different (**a**,**b**) operation temperatures and (**c**,**d**) different inlet flow rates at 240 °C.

**Figure 5 micromachines-14-01845-f005:**
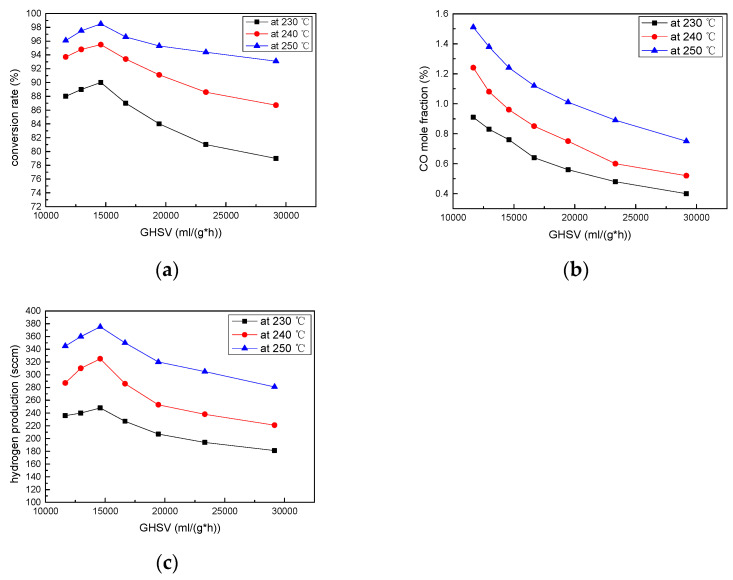
(**a**) Methanol conversion rate (**b**) CO mole fraction (**c**) hydrogen productivity of Swiss-Roll reformers at different GHSVs.

**Figure 6 micromachines-14-01845-f006:**
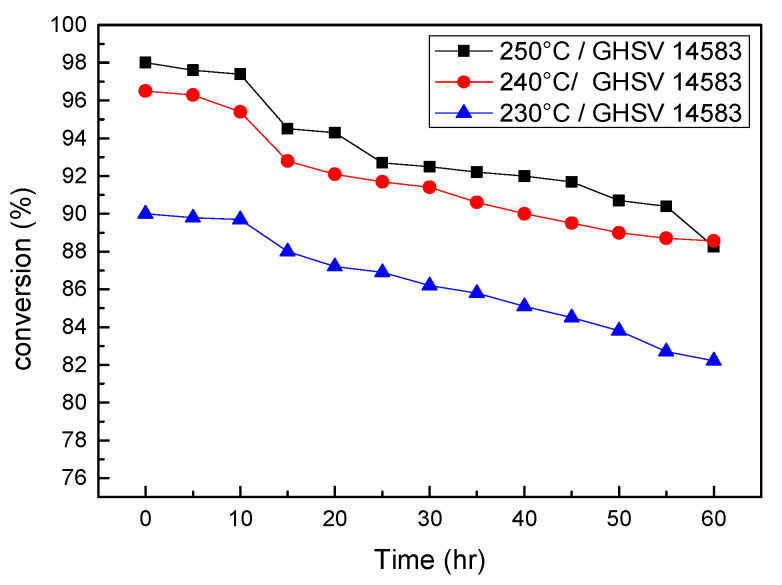
Conversion rate variations for various operation temperatures with the same GHSV in long-term durability test.

**Figure 7 micromachines-14-01845-f007:**
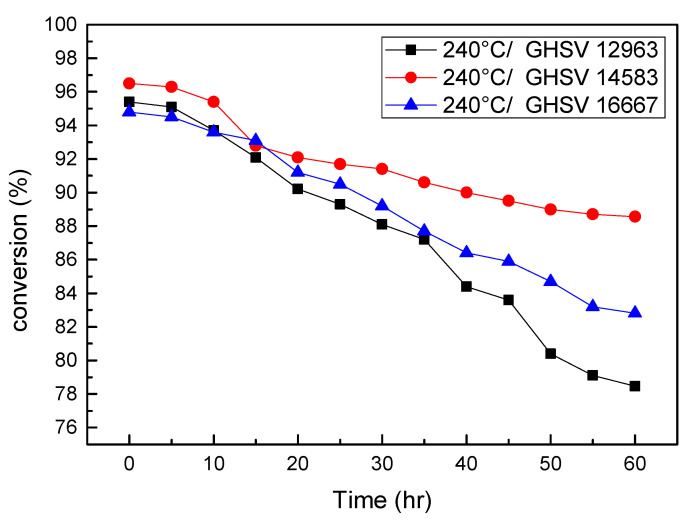
Conversion rate variations for various GHSVs of the reformer operated at the same temperature in long-term durability test.

**Table 1 micromachines-14-01845-t001:** The effect of temperature with the same GHSV in long-term durability test.

GHSV 14,583 (mL/(g*h))	250 °C	240 °C	230 °C
Decay rate (%/h)	0.1623	0.1322	0.1297
Final conversion rate (%)	88.66	88.12	84.69

**Table 2 micromachines-14-01845-t002:** The effect of GHSV with the same temperature in long-term durability test.

Temperature 240 °C	12,963 (mL/(g*h))	14,583 (mL/(g*h))	16,667 (mL/(g*h))
Decay rate (%/h)	0.2821	0.1322	0.1998
Final conversion rate (%)	78.474	88.568	82.52

## Data Availability

The data presented in this study are available upon request.

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
