# Peer review of "A Swiss-Roll-Type Methanol Mini-Steam Reformer for Hydrogen Generation with High Efficiency and Long-Term Durability"

_micromachines, 2023, doi:10.3390/mi14101845_

Round 1

Reviewer 1 Report

This paper discusses the effect of reformer structure on the methanol steam reforming reaction. It proposes a novel Swiss roll-type reactor, which is an interesting approach in terms of temperature uniformity and flow uniformity compared to the conventional straight-type fixed bed reactors.

On the other hand, it is mentioned that the catalyst durability was greatly improved compared to the reference [29], but the cause of this improvement should be discussed in more detail. In the introduction section, it was mentioned that the Swiss roll-type reactor has a uniform temperature and no hot spots.

Measurement of the temperature distribution in the reactor is essentially necessary, but if this is difficult, it should be clarified from the analysis of the catalyst after the reaction whether non-uniform sintering of Cu components, etc. were observed.

Also, the cause of the maximum value at GHSV 15000 in Fig. 4 should be discussed.

Reviewer 2 Report

The work is on the intensification of steam methanol reforming along with providing a design for compact types. Also, it discusses the thermal management of the reactor, which is one of the most significant aspects in the reforming reaction. The work is very interesting, however, some revision is required for further consideration by Micromachines.

*In the abstract, the authors claimed that they have achieved better thermal efficiency "compared with other geometries." The authors have to provide more details such as what geometries and how is this improved in a quantitative value. Also, the comparison should be revealed in the results. Also the comparison should be in the same operating conditions supported with results or with referenced results. The geometries the authors compared their results with have to be revealed explicitly.

 *In page 2, the authors mention that Cu, Zn and Al are mostly used for catalysts preparation. However, they are usually used as a "support", thus, the authors have to specify this issue. Also, the use of Ni is the most common for reforming, nevertheless, the authors have not mentioned it. 

*References [20, 21] are not well placed

*The authors have claimed that their study focuses on the thermal performance of the reformer. Thus, mentioning the current updates and solutions for thermal performance have to be reviewed from literature. The following references are suggested:  | doi.org/10.1016/j.applthermaleng.2022.119140 | doi.org/10.1016/j.ijhydene.2021.12.250  doi.org/10.1016/j.powtec.2023.118664 | doi.org/10.1016/j.ces.2023.118987|  doi.org/10.1016/j.ijhydene.2020.07.182  |  doi.org/10.1016/j.tsep.2023.101868

*Provide the dimension details in Fig. 1a. You have provided the details for Fig. 1b in Fig 2, but no details given for Fig. 1a.

*The acknowledgment involves the use of Ansys Fluent. However, it was not provided in the manuscript. The authors have to mention where it was used in their paper in the method parts. 

*The authors have to emphasize the manuscript novelty in detail.

Round 2

Reviewer 1 Report

In this revised version, the authors have accurately addressed the reviewers' comments, and I consider the manuscript to be worthy of acceptance.

Reviewer 2 Report

A further improvement of the manuscript has to be considered. The numerical model has to be effectively portrayed, in the following comments. 

* For Fig. S1, the authors could involve it in the main manuscript. Also, the main species' contour should be added.

* In Table S1, I recommend naming the pressure drop instead of the difference.

* In Fig. S2c, the authors could add the numbers in digital form with the unit. Also, the contour and the values have to be split (Fig. S2c and Fig. S2d as they are not dependent. 

*How can the authors explain the low-temperature difference despite the endothermal reaction?

* In the introduction, the authors should sufficiently discuss thermal performance background and updates in literature. The following research is suggested: doi.org/10.1016/j.applthermaleng.2022.119140 , doi.org/10.1016/j.ijhydene.2021.12.250  doi.org/10.1016/j.powtec.2023.118664 , doi.org/10.1016/j.ces.2023.118987 ,  doi.org/10.1016/j.ijhydene.2020.07.182  ,  doi.org/10.1016/j.tsep.2023.101868

*The computational fluid dynamic details such as the boundary conditions and governing equations should be added. Also, the mesh figures should be provided before the results. 

Round 3

Reviewer 2 Report

The authors have revised the manuscript. However, there are significant concerns raised. 

*The authors must conveniently present the boundary conditions and governing equations. Many inaccuracies were found. For example, and not limited to:

-The catalyst porosity is not a CFD boundary condition.

-The provided Governing equation is only a source term

The authors must seek to accurately improve the portrayed CFD details to avoid serious non-professional writing. I suggest the following reference on CFD modeling to correct the model description: doi.org/10.1016/j.ijhydene.2018.12.203
